# UID-Dual Transcriptome Sequencing Analysis of the Molecular Interactions between *Streptococcus agalactiae ATCC 27956* and Mammary Epithelial Cells

**DOI:** 10.3390/ani14172587

**Published:** 2024-09-05

**Authors:** Jishang Gong, Taotao Li, Yuanfei Li, Xinwei Xiong, Jiguo Xu, Xuewen Chai, Youji Ma

**Affiliations:** 1College of Science and Technology, Gansu Agriculture University, Lanzhou 730070, China; gongjishang@126.com (J.G.); ttli2018@163.com (T.L.); 2Institute of Biological Technology, Nanchang Normal University, Nanchang 330030, China; li-yuan-fei@outlook.com (Y.L.); xinweixiong@hotmail.com (X.X.); xujiguo@ncnu.edu.cn (J.X.); chaixuewen03@ncnu.edu.cn (X.C.)

**Keywords:** mastitis, *Streptococcus agalactiae*, UID-Dual, transcriptome sequencing, mammary epithelial cells

## Abstract

**Simple Summary:**

The prevention and control of subclinical mastitis in dairy cows remains challenging. The pathogen *Streptococcus agalactiae ATCC 27956* is a major Gram-positive bacterium that can damage host cells by infecting the mammary glands of cows. To analyze the molecular interactions during *Streptococcus agalactiae* infection, UID-Dual transcriptome sequencing was performed, and bioinformatics tools were used for analysis. Differentially expressed genes were mainly enriched in biological processes related to inflammation, immune response, and cancer. *Streptococcus agalactiae* can express genes that interfere with lncRNA in mammary epithelial cells, indirectly affecting the alternative splicing of lncRNA target genes and thus influencing normal cellular processes. This study provides potential therapeutic targets for the prevention and treatment of subclinical mastitis caused by *Streptococcus agalactiae*.

**Abstract:**

*Streptococcus agalactiae ATCC 27956* is a highly contagious Gram-positive bacterium that causes mastitis, has a high infectivity for mammary epithelial cells, and becomes challenging to treat. However, the molecular interactions between it and mammary epithelial cells remain poorly understood. This study analyzed differential gene expression in mammary epithelial cells with varying levels of *S. agalactiae* infection using UID-Dual transcriptome sequencing and bioinformatics tools. This study identified 211 differentially expressed mRNAs (DEmRNAs) and 452 differentially expressed lncRNAs (DElncRNAs) in host cells, primarily enriched in anti-inflammatory responses, immune responses, and cancer-related processes. Additionally, 854 pathogen differentially expressed mRNAs (pDEmRNAs) were identified, mainly enriched in protein metabolism, gene expression, and biosynthesis processes. Mammary epithelial cells activate pathways, such as the ERK1/2 pathway, to produce reactive oxygen species (ROS) to eliminate bacteria. The bacteria disrupt the host’s innate immune mechanisms by interfering with the alternative splicing processes of mammary epithelial cells. Specifically, the bacterial genes of *tsf*, *prfB*, and *infC* can interfere with lncRNAs targeting *RUNX1* and *BCL2L11* in mammary epithelial cells, affecting the alternative splicing of target genes and altering normal molecular regulation.

## 1. Introduction

Udders are vital functional organs in dairy cows and are a significant source of protein (milk) for humans. Milk is mainly stored primarily in the alveoli between milkings, and oxytocin release is essential for “squeezing” the alveoli and milk dripping [1]. Typically, after the mammary gland is infected with pathogenic microorganisms or stimulated by physical, chemical, or other factors, inflammatory changes in the plasma or parenchymal tissue of the mammary gland can lead to mastitis. Mastitis syndrome is caused by a variety of microorganisms, predominantly bacteria, that are divided into contagious and environmental types. The sources of contagious bacteria are infected quarters and cows, whereas the source of environmental bacteria is the environment [2]. Based on clinical signs, mastitis can be divided into clinical and subclinical forms [3,4]. Clinical mastitis is characterized by pronounced signs of inflammation in the udder, the presence of microbials, and changes in the chemical properties of milk [5]. Cows with clinical mastitis are infected with pathogenic bacteria that cause fibrosis and atrophy of the mammary glands, leading to premature culling [6]. However, dairy cows with subclinical mastitis exhibit only mild inflammation after pathogenic bacteria enter the teat ducts or papillae. The somatic cell count (SCC) in milk increases without clinical manifestations [7]. Although subclinical mastitis does not pose an immediate risk, if left untreated, it may eventually turn into clinical mastitis, resulting in significant economic losses.

A study by Paramanandham et al. indicated that *Staphylococcus aureus* is the most common contagious pathogen, *Escherichia coli* is the main pathogen causing clinical mastitis, and *Streptococcus* spp. cause both subclinical and clinical mastitis worldwide [8]. In many cases of subclinical mastitis, the inflammatory response may be predominantly caused by *Streptococcus agalactiae* [9,10,11]. *S. agalactiae* is classified as group B according to the Lancefield bacterial taxonomy. The bacterium is mostly β-hemolytic, although some non-hemolytic and CAMP-positive strains have been observed [12]. Owing to its variety of virulence factors, *S. agalactiae* can adhere to and invade host cells, inducing an inflammatory response in the mammary glands. Therefore, it is essential to analyze the transcriptional regulation of inflammatory genes during invasion by *S. agalactiae* into the mammary glands. The mammary epithelium is the first line of defense in the mammary glands. It can effectively initiate an immune response by eliminating pathogens before abnormal changes occur in the mammary glands, which is crucial for resistance to mastitis and affects susceptibility [13]. When a pathogen successfully infringes on the host’s physical defenses, the mammary gland epithelial detects bacteria through specific pattern recognition receptors and initiates a series of immune responses. Currently, researchers have conducted high-throughput sequencing studies in mammary cells infected with *S. agalactiae* and have identified several immune-related receptors or pathways; however, the findings are inconsistent. The mechanisms underlying *S. agalactiae* infections in subclinical mastitis are still poorly understood.

Zhang et al. found that, compared to healthy cow mammary glands, 129 differentially expressed genes and 144 differentially expressed proteins were identified in mammary glands infected with *S. agalactiae* [14]. Intramammary infection with *S. agalactiae* triggers a complex host-innate immune response that involves complement and coagulation cascades, ECM–receptor interaction, focal adhesion, phagosomes, and bacterial invasion of epithelial cell pathways. Tong et al. used proteomics to discover that the differentially expressed proteins included enzymes and proteins associated with various metabolic processes and cellular immunity in *S. agalactiae*-infected bovine mammary epithelial cells [15]. Subsequently, they used ubiquitinome analysis to determine whether ubiquitinated proteins were associated with the regulation of cell junctions in the host [16]. Sbardella et al. exogenously infected dairy cow mammary glands with *S. agalactiae* and identified 122 differentially expressed genes from sequencing data based on three different statistical methods; however, only the platelet activation pathway showed a significant enrichment [17]. Furthermore, differentially expressed genes have been identified in mammary alveolar tissue infected with *S. agalactiae* that are mainly involved in the innate immune response, the inflammatory response, the signaling of chemokine, Wnt signaling, and in complement and coagulation cascades compared with normal tissues [18]. Richards et al. concluded that lactose metabolism is an important metabolic pathway for *S. agalactiae* to adapt to the bovine mammary environment, as determined by sequence analysis of isolated *S. agalactiae* [19]. In mammary glands infected with *S. agalactiae*, differentially expressed miRNAs are mainly involved in the signaling of RIG-I-like receptors, the detection of cytosolic DNA, and the Notch signaling pathways [20].

Most studies have been conducted in vivo, and research on the immunological changes following mammary epithelial cell infection is lacking. Therefore, studying immune regulation in mammary epithelial cells infected with *S. agalactiae* is essential. The transcriptome serves as a powerful indicator of the physiological state of a cell (healthy or diseased). Consequently, transcriptome analysis has become a crucial tool for understanding the molecular changes that occur during bacterial infections in eukaryotic cells. Previously, transcriptomic studies were limited to the analysis of mRNA expression in bacterial pathogens or infected eukaryotic host cells. However, the increasing sensitivity of high-throughput RNA sequencing now enables “UID-Dual RNA transcriptome sequencing” studies, simultaneously capturing all classes of coding and non-coding transcripts in both the pathogen and host [21].

To the best of our knowledge, this is the first report of UID-Dual RNA transcriptome sequencing in mammary epithelial cells infected with *S. agalactiae*. Identifying key differentially expressed genes and pathways in infected mammary epithelial cells provides a basis for a better understanding of the central mechanisms of host defense during subclinical infections, such as mastitis.

## 2. Materials and Methods

### 2.1. Bacterial Strains and Growth Conditions

*S. agalactiae* strain *ATCC 27956* was inoculated onto Edwards Medium Modified (EMM, Hope Bio-Technology Co., Ltd., Qingdao, China) Agar and incubated at 37 °C for 24 h. A single colony was randomly selected and cultured in Todd Hewitt Broth (THB, Hope Bio-Technology Co.) with agitation at 37 °C for 12 h, and the growth was monitored by measuring the OD600 nm.

### 2.2. Cell Culture

MAC-T cells were cultured in T25 cell culture flasks with Dulbecco’s modified eagle culture medium (DMEM, Gibco, Grand Island, NY, USA) containing 10% fetal bovine serum (FBS, Gibco) and maintained in 5% CO_2_ at 37 °C. The cells were cultured until they reached 80% of the confluence for further experiments.

### 2.3. Intracellular Infection Model

The intracellular infection model followed the method described by Tong et al. [16]. MAC-T cells were cultured in T75 cell culture flasks until they reached a density of 1 × 10^6^ cells/mL. Control (M Group) and MOI (100:1) groups were then established. The MOI group was incubated for 2 h (S Group) and 6 h (H Group), respectively. Each group included at least three biological replicates. The cells were washed twice with phosphate-buffered saline (PBS), and DMEM containing lysozyme (20 μg/mL) and gentamicin (100 μg/mL) were added. The culture was maintained at 37 °C in 5% CO_2_ for 2 h to remove the extracellular bacteria. The collected sterilized cell culture medium was applied to bacterial plates to ensure the elimination of extracellular bacteria. The MAC-T cells were rinsed three more times with PBS to remove any remaining extracellular adherent bacteria and were then cultured in 10% FBS-DMEM.

### 2.4. RNA Extraction and cDNA Library Construction

Eukaryotic RNA contains a polyA tail, whereas prokaryotic RNA lacks this feature. Therefore, using polyA capture can only obtain expression information from eukaryotes, leading to the loss of expression data from prokaryotes. To address this, such interaction protocols use rRNA depletion, removing rRNA from both eukaryotes and prokaryotes. This is followed by library construction, sequencing, and analysis to obtain gene expression profiles of both the pathogen and the host. Before library amplification, each reverse-transcribed cDNA fragment is tagged with a Unique Identifier (UMI), also known as a digital tag. This tag accompanies the fragment throughout amplification, sequencing, and analysis. After UMI sequencing library construction, all PCR-amplified products from the same fragment carry the same digital tag. Upon sequencing, UMIs are used to trace the origin of each fragment, allowing for the merging of fragments with the same sequence and UMI, thereby accurately removing PCR duplicates and restoring the original state of the sample before amplification. During this process, PCR amplification and sequencing errors can also be corrected: errors will result in the same UMI corresponding to multiple different sequences, which can be corrected by comparing the similarity of these sequences. This sequencing method is known as UID-Dual transcriptome sequencing.

Total RNAs were extracted from both the control and *S. agalactiae*-treated group using TRIzol Reagent (Invitrogen, Carlsbad, CA, USA) following the manufacturer’s instructions. DNA digestion was performed after RNA extraction using DNaseI. RNA quality was assessed by measuring the A260/A280 ratio using a Nanodrop^TM^ OneC spectrophotometer (Thermo Fisher Scientific Inc., Waltham, MA, USA). RNA integrity was confirmed using a Qsep100 (BiOptic Inc., Changzhou, China) and a 5300 Fragment Analyzer system (Agilent, Santa Clara, CA, USA). The qualified RNAs were quantified using a Qubit3.0 with the Qubit^TM^ RNA Broad Range Assay kit (Life Technologies, Carlsbad, CA, USA, Q10210).

A total of 2 μg total RNA was used for stranded RNA sequencing library preparation using a KC-Digital^TM^ Total RNA Library Prep Kit (Wuhan Seqhealth Co., Ltd., Wuhan, China), Ribo-off rRNA Depletion Kit (Vazyme, Nanjing, China), MICROB Express Kit (Thermo), and Ribo-off rRNA Depletion Kit (Bacteria), (Vazyme) following the manufacturer’s instruction. The library products, ranging from 200 to 500 bps were enriched, quantified, and sequenced using a DNBSEQ-T7 sequencer (MGI Tech Co., Ltd., Shenzhen, China) with PE150 mode. The UID-Dual RNA transcriptome sequencing experiment, high-throughput sequencing, and data analysis were conducted by Seqhealth Technology Co., Ltd. (Wuhan, China).

### 2.5. RNA Transcriptome Sequencing Data Analysis

Raw sequencing data were first filtered by using fastp (version 0.23.0), low-quality reads were discarded, and the reads contaminated with adaptor sequences were trimmed. Clean reads were further processed using in-house scripts to eliminate duplication bias introduced during library preparation and sequencing. In brief, clean reads were first clustered based on UMI sequences, with reads sharing the same UMI sequence grouped into the same cluster. Reads within the same cluster were compared by pairwise alignment, and those with sequence identity exceeding 95% were assigned to a new sub-cluster. After all sub-clusters were generated, multiple sequence alignment was performed to obtain one consensus sequence for each sub-cluster. Following these steps, any errors and biases introduced during PCR amplification or sequencing were removed. 

### 2.6. Reads Alignment and Differential Expression Analysis of RNA Transcriptome Sequencing

The reference genomes of the two species were merged, and then the deduplicated data were mapped to the reference genomes of *Bos taurus* from http://asia.ensembl.org/Bos_taurus/Info/Index (accessed on 28 May 2023) and *S. agalactiae* from https://bacteria.ensembl.org/Streptococcus_agalactiae_gca_900458965/Info/Index (accessed on 28 May 2023) using STAR software (version 2.5.3a) with default parameters. The reads mapped to the exon regions of each gene were counted using feature counts (Subread-1.5.1; Bioconductor), and then RPKM was calculated.

Differentially expressed genes between groups were identified using the edgeR package (version 3.28.1). A *p*-value cutoff of 0.05 and a fold-change cutoff of 1 or −1 were used to determine the statistical significance of gene expression differences. 

### 2.7. Bioinformatics Analysis of Differentially Expressed Genes

The differentially expressed genes (DEmRNA) obtained were subjected to Gene Ontology (GO) and Kyoto Encyclopedia of Genes and Genomes (KEGG) pathway enrichment analysis. Commonly enriched GO terms and KEGG pathways across the three comparison groups were statistically analyzed. GO and KEGG analysis was performed using the DAVID 2021 (December 2021) functional annotation tool. The alternative splicing prediction was predicted by rMATS (Version 3.2.5) [22]. The differentially expressed genes involved in alternative splicing across the three comparison groups were further analyzed for GO and KEGG enrichment, and common enrichments were identified. The differentially expressed long non-coding RNAs (DElncRNA) were predicted using four programs: CPC (Version beta), CPAT (Version 1.2.4), CNCI (Version 2), and Pfam (Version 27.0) [23,24,25,26]. The significant DElncRNAs were subjected to target gene prediction using RIsearch (Version 2.0) [27]. A Venn analysis was then performed on the DEmRNA, differentially spliced genes, and DElncRNA across the three comparison groups to identify candidate genes and their targeted lncRNAs. To gain deeper insights into the expression patterns of candidate genes, Mfuzz (version 2.64.0) was used to identify potential time-series patterns and cluster genes with similar expression profiles [28]. Subsequently, the bacterial infection data from two groups were analyzed for differentially expressed bacterial genes (pDEmRNA), with significant genes being filtered and subjected to GO and KEGG enrichment analysis. Finally, gene co-expression analysis was conducted on the target genes, targeted lncRNAs, and differentially expressed bacterial genes. Co-expression analysis was performed at https://www.bioinformatics.com.cn (last accessed on 20 June 2024), an online platform for data analysis and visualization [28].

## 3. Results

### 3.1. Transcriptome Assembly Profiles Evaluation

Thirteen samples were sequenced, yielding 156.06 Gb of mRNA and lncRNA transcription data (Table 1). Raw reads were filtered, and clean data were analyzed downstream.

### 3.2. Analysis of Differentially Expressed mRNAs

The cluster pattern analysis of host DEmRNAs between the control (*n* = 5), *S. agalactiae*—S groups (*n* = 5), and *S. agalactiae*—H groups (*n* = 3) is illustrated in Figure 1A. A total of 3370 DEmRNAs (S_M) (Appendix A) were filtered using the thresholds of *p* < 0.05 and |log2(fold-change)| > 1, which revealed 2001 upregulated and 1369 downregulated genes (Figure 1B). A total of 4730 DEmRNA (H_M) were identified, of which 2472 were upregulated and 2258 were downregulated (Figure 1C) (Appendix A). A total of 2085 DEmRNA (H_S) were identified, of which 1102 were upregulated and 983 were downregulated (Figure 1D) (Appendix A).

Gene Ontology (GO) was used to classify the functions of DEGs (Figure 2). The DEGs enriched in the three comparison groups were annotated using three categories of GO: biological processes (BPs), cellular components (CCs), and molecular functions (MFs).

In the comparisons of S_M, H_M, and H_S, the GO enrichment analysis showed that the upregulated genes in BPs were mainly enriched in negative chemotaxis, the extrinsic apoptotic signaling pathway in the absence of ligand, small GTPase-mediated signal transduction, and regulation of ERK1 and ERK2 cascade. In terms of MFs, these were primarily enriched in transcription factors, whereas the CCs were primarily enriched in cell membrane structures and complexes. Enriched downregulated genes were mainly involved in NADH dehydrogenase (ubiquinone) activity and translation elongation factor activity in MFs, in the Lsm2-8 complex and nucleolus in CCs, and in mRNA splicing via the spliceosome and translational elongation in BPs. Furthermore, these genes were also enriched in apoptosis-related pathways.

Enrichment analysis of the KEGG pathway for the DEmRNAs is shown in Figure 2C. In the three comparison groups, the signaling pathways were primarily enriched in disease-related pathways such as cancer, leukemia, and diabetes. Additionally, they were primarily enriched in environmental information processing pathways, such as the MAPK signaling pathway, the TGF-beta signaling pathway, and the Notch signaling pathway. The pathways involved in cellular processes included the p53 signaling pathway, the apoptotic signaling pathway, and cellular senescence. Pathways related to organismal systems included thermogenesis and the IL-17 signaling pathway.

### 3.3. Analysis of Host mRNA Alternative Splicing

Using rMATs to analyze differential alternative splicing (AS) events in S_M, H_M, and H_S, 130,178 alternative splicing events were detected using the target and junction reads. After setting the threshold *p*-value < 0.05 and |Δψ| > 0.1 for alternative splicing filtering, a total of 10,750 differentially expressed alternative splicing events were identified in the three comparison groups (Table 2).

Of the three comparison groups, the SE type was the most frequently identified, with a total of 104,687 events, followed by the MXE type with 21,901 events. The A5SS, A3SS, and RI types were relatively less frequent, with 982, 1216, and 1392 events, respectively. Differential analysis of AS events revealed 3980 differentially expressed SE types across the three comparison groups, of which 1505 were upregulated and 2475 were downregulated. The analysis identified 6501 differentially expressed MXE types, with 3501 upregulated and 3000 downregulated. 

In the S_M group, the numbers of differentially spliced genes identified in the SE, MXE, A5SS, A3SS, and RI events were 983, 2065, 32, 25, and 24, respectively. The differentially spliced genes in the H_M group were 1684, 2505, 44, 32, and 45, respectively. In the H_S group, the differentially spliced genes were 1313, 1931, 26, 21, and 20. Among the differentially spliced genes, the MXE type was the most common, while the A3SS type was the least.

Differentially spliced genes in the three comparison groups were mainly enriched in BPs, including catabolic process dependent on ubiquitin, positive regulation of GTPase activity, and polyubiquitination of proteins. CCs showed primary enrichment in the nucleus, cytoplasm, nucleoplasm, and nuclear body. MFs were predominantly associated with ATP binding, RNA binding, and GTPase activator activity (Figure 3).

Enrichment analysis of the KEGG pathway in AS DEmRNAs is shown in Figure 4. In the three comparison groups, signaling pathways were enriched in various categories, including disease, metabolism, genetic information processing, environmental information processing, and cellular processes. Metabolic pathways included those involved in lysine degradation and fatty acid metabolism. Pathways involved in the processing of genetic information include ubiquitin-mediated proteolysis and nucleotide excision repair. Sphingolipid signaling is the main pathway enriched in environmental information processing. Pathways enriched in cellular processes included those involved in focal adhesions, tight junctions, and autophagy. Disease-related pathways included those associated with *Yersinia* infection, renal cell carcinoma, and bacterial invasion of epithelial cells.

### 3.4. Analysis of Differentially Expressed lncRNA

Each sample yielded novel DElncRNAs (Figure 5A), and according to the CNCI, COC, Pfam, and CPAT programs, 5995 novel DElncRNAs were identified (Figure 5B). After correction, the uniformity of the samples was relatively high. A heatmap showed the hierarchical clustering of DElncRNAs (Figure 5C). A volcano plot illustrating the DElncRNAs between S_M, H_M, and H_S is shown in Figure 5D–F. The S_M comparison included 4377 significantly different genes, with a total of 2798 targeted genes. The H_M comparison revealed 4050 significantly different genes, with 376 targeted genes. The H_S comparison included 1868 significantly different genes, with a total of 189 targeted genes. In total, 452 mRNAs targeted by lncRNAs were identified as common among the three comparison groups in the final prediction, such as *RUNX1* was targeted by *TCONS_00001766* and *TCONS_00001789*, *ENSBTAG00000048558* targeted to *DNAH12*, *TCONS_00026618* targeted to *TCHP*, *TCONS_00085732* targeted to *IDS*, *TCONS_00019399* targeted to *ARRB1*, *TCONS_00065698* targeted to *LSMEM1*, *RNase-MRP* targeted to *TPM2*, and *TCONS_00007590* targeted to *BCL2L11* (Figure 6) (Appendix A).

The Venn analysis revealed 211 DEmRNAs among the three comparative groups. Eight genes were common among differentially expressed genes, alternatively spliced genes, and lncRNA-targeted mRNAs in the three control groups, namely *TPM2*, *ARRB1*, *TCHP*, *BCL2L11*, *LSMEM1*, *RUNX1*, *DNAH12*, and *IDS* (Figure 7) (Appendix A).

To better understand the dynamic changes in gene expression during metastatic progression, we classified all DEmRNAs into eight patterns (Cluster 1, …, Cluster 8) using Mfuzz (Figure 8).

Genes in Cluster 1 were upregulated in two stages in the S and H groups after cells were infected, while genes in Cluster 3 did not show significant differences between the M and S groups but were rapidly upregulated in the H group. The genes in Clusters 2 and 4 were upregulated in the S group and downregulated in the H group; however, the difference between the M and S groups was not significant in Cluster 2. The genes in Cluster 4 were downregulated in the H group, but their expression levels were still higher than those in the M group. The genes in Clusters 5 and 8 were negatively regulated in the S and M groups; however, the genes in Cluster 8 showed significant differences between the genomes, with no significant difference between the S and M groups. Genes in Cluster 7 were downregulated in the S group and rapidly increased in the H group, with significant differences between the H group and both the M and S groups. By examining each cluster, *TPM2* and *TCHP* were found to be enriched in Cluster 5; *ARRB1*, *BCL2L11*, *LSMEM1*, and *IDS* in Cluster 3; and *RUNX1* and *DNAH12* in Cluster 1 (Figure 9).

Time-series analysis indicated that Clusters 1, 3, and 6 exhibited an upregulated trend in gene expression, whereas Clusters 5 and 8 exhibited a downward trend. GO enrichment analysis of the five clusters revealed that in Cluster 1, *RUNX1* and other genes were enriched in BPs involving positive regulation of transcription by RNA polymerase II (Appendix A). Additionally, this cluster was enriched both in the apoptotic process and negative regulation of apoptosis. *RUNX1* and other genes were associated with the CCs of the nucleus and nucleoplasm. Regarding MFs, *RUNX1* in Cluster 1, along with other genes, was enriched in ATP and DNA-binding transcription activator activities involving RNA polymerase. *RUNX1* was involved in implicated in chronic myeloid leukemia and transcriptional misregulation of cancer signaling pathways.

Cluster 3 (Appendix A) mainly participated in BPs, including the inflammatory response, immune response, innate immune response, apoptotic process, positive regulation of the ERK1 and ERK2 cascade, and regulation of reactive oxygen species (ROS) metabolism. *BCL2L11* and other genes were enriched in the apoptotic process and in the positive regulation of apoptosis. They were also associated with CCs of the membrane and mitochondria. With regard to MFs, Cluster 3 was mainly enriched in transcriptional activator activity, RNA polymerase regulatory region sequence-specific binding, GTPase activator activity, and ubiquitin-protein ligase activity. KEGG analysis indicated that *BCL2L11* participated primarily in pathways related to cancer, Epstein–Barr virus infection, FoxO signaling, and apoptosis, and most genes were involved in the MAPK signaling pathway, metabolic pathways, IL-17 signaling pathway, NOD-like receptor signaling pathway, and TNF signaling pathway.

Cluster 6 (Appendix A) mainly participated in BPs, including the regulation of transcription by RNA polymerase II, the negative regulation of Rho protein signal transduction, the catabolic process of ubiquitin-dependent proteins, phosphorylation, signal transduction, and small GTPase-mediated signal transduction. For the MFs, most genes in Cluster 6 were involved in GTPase activator activity, ubiquitin-protein transferase activity, ubiquitin-protein ligase activity, ATP binding, protein binding, RNA polymerase II cis-regulatory region, and DNA-binding transcription activator activity. The KEGG pathways mainly involved the MAPK signaling pathway, transcriptional misregulation in cancer, and other cancer pathways.

Cluster 5 (Appendix A) was mainly enriched in the BPs of RNA splicing via the spliceosome and protein folding. CCs were enriched in the spliceosomal complex. The MFs were enriched in RNA binding, unfolded protein binding, and transcription coactivator activity. KEGG analysis revealed that these genes were involved in the spliceosome and nucleotide excision repair pathways. Cluster 8 (Appendix A) shared similar enrichment with Cluster 5, including RNA splicing via the spliceosome and protein folding. Furthermore, it was enriched in the regulation of transcription by RNA polymerase II and immune system processes. Similar to Cluster 5, this cluster was enriched in the spliceosomal complex for the CCs; however, most genes were also enriched in the nucleus and cytoplasm. For MF, in addition to being enriched in RNA binding like Cluster 5, it was also enriched in DNA-binding transcription factor activity specific to RNA polymerase II, RNA polymerase II cis-regulatory region sequence-specific DNA binding, and DNA-binding transcription repressor activity specific to RNA polymerase II.

### 3.5. Analysis of pDEmRNAs of S. agalactiae ATCC 27956

The cluster pattern analysis of pathogenic pDEmRNAs between normally treated groups of *S. agalactiae ATCC 27956* (*n* = 5) and the deeply treated groups of *S. agalactiae* (*n* = 3) is shown in Figure 9A. A total of 864 pDEmRNAs (Appendix A) were filtered using thresholds of *p*-values < 0.05 and log2(fold-change) > 1 or <−1, of which 861 were upregulated and 3 were downregulated (Figure 9B).

GO enrichment analysis of pDEmRNAs revealed that enriched BPs were involved mainly in protein metabolism, gene expression, and biosynthesis, including the organic substance biosynthetic process, cellular biosynthetic process, regulation of transcription, cellular protein metabolic process, peptide metabolic process, and gene expression (Figure 9D). The enriched CCs were located mainly in the cell, cell part, intracellular, intracellular part, cytoplasm, and protein-containing complex. The enriched MFs mainly included RNA binding, catalytic activity, transferase activity, and oxidoreductase activity. The KEGG pathway analysis showed that the main enrichments are in metabolism-related pathways (Figure 9C), such as the biosynthesis of amino acids, purine metabolism, glycolysis/gluconeogenesis, carbon metabolism, and fatty acid metabolism. Further enrichments were observed in the genetic information processing pathways, mainly ribosome, aminoacyl-tRNA biosynthesis, and protein export. The pathways related to environmental information processing include ABC transporters, the phosphotransferase system, and the two-component system. Disease-related signaling pathways included vancomycin resistance, beta-lactam resistance, and cationic antimicrobial peptide (CAMP) resistance. The D-alanine metabolism signaling pathway was involved in the construction of peptidoglycans from the cell wall and teichoic acid.

### 3.6. Gene Co-Expression Analysis Interaction Network

Using gene co-expression analysis, which targeted the lncRNAs of the eight candidate genes in Clusters 1, 3, and 5, as well as differentially expressed genes related to the transcriptional regulation of *S. agalactiae ATCC 27956*, six upregulated genes of *S. agalactiae* were correlated with *TCONS_00001766-RUNX1*, *TCONS_00001789-RUNX1*, *ENSBTAG00000048558-DNAH12*, *TCONS_00026618-TCHP*, *TCONS_00085732-IDS*, *TCONS_00019399-ARRB1*, *TCONS_00065698-LSMEM1*, *RNase-MRP-TPM2*, and *TCONS-00007590-BCL2L11*. *tsf*, *prfB*, and *infC* significantly affected the nine candidate lncRNAs (Figure 10).

### 3.7. The Expression Level of Candidate Genes

Statistical analysis of high-throughput sequencing data showed that *TPM2*, *TCHP*, and *TCONS_00026618* were downregulated, whereas *LSMEM1*, *RUNX1*, *IDS*, *DNAH12*, *ARRB1*, *BCL2L11*, and lncRNAs *RNase_MRP*, *TCONS_00001766*, *TCONS_00065689*, *TCONS_00001789*, *TCONS_00085732*, *TCONS_00019399*, *TCONS_00007590*, and *ENSBTAG00000048558* were upregulated (Figure 11).

## 4. Discussion

*S. agalactiae ATCC 27956*, isolated from bovine udder infections and commonly referred to as Group B Streptococcus (GBS), is a zoonotic pathogen and a highly infectious Gram-positive bacterium [12]. The most common form of *S. agalactiae* mastitis is the chronic subclinical form, which leads to fibrosis of the mammary gland and loss of productivity and may progress to clinical mastitis if left unchecked [12]. Research has demonstrated that *S. agalactiae* infection of the mammary glands initiates a series of innate immune responses in the host. During this time, bacteria invade mammary epithelial cells to evade host defenses and antibiotics [14]. Various molecular interactions occur during this infection process, highlighting the importance of studying the molecular changes that mediate mastitis.

Zhang et al. used transcriptomics and proteomics to analyze mammary tissues infected with *S. agalactiae* and investigated the host’s immune response to the pathogen [14]. Tong et al. conducted ubiquitination sequencing and analysis of mammary epithelial cells infected with *S. agalactiae* [16]. Mayara et al. perfused mammary tissues with *S. agalactiae* to observe transcriptional changes and focused on the most affected biological functions and pathways [18]. These studies mainly examined the resistance mechanisms of mammary tissues or the ubiquitination processes in mammary epithelial cells after infection by *S. agalactiae*, each focusing on different aspects. Although subclinical mastitis in dairy cows is caused by a combination of multiple factors, studying a single strain may not be fully representative and has certain limitations. Focusing on a single strain allows for a detailed understanding of its biological characteristics, pathogenic pathways, and infection mechanisms, providing a theoretical foundation for developing targeted treatments and control measures. In this study, we used mammary epithelial cells in vitro infected with *S. agalactiae ATCC 27956* for varying times and performed an interaction transcriptome analysis to investigate the molecular mechanisms of host–pathogen interactions during transcriptional regulation.

### 4.1. Biological Function of Mammary Epithelial Cells Undergo Significant Changes after Being Infection by S. agalactiae ATCC 27956

GO enrichment analysis of DEmRNAs revealed enrichment in processes related to inflammation, disease occurrence, damage repair, and regulation of apoptosis. Upregulated genes were enriched in exogenous apoptosis regulation processes, whereas downregulated genes were enriched in endogenous apoptosis signaling pathways. This could be due to the induction of exogenous apoptotic signals and the inhibition of endogenous apoptotic processes after pathogen invasion. Upregulated genes were significantly enriched in small GTPase-mediated signal transduction and regulation of the ERK1 and ERK2 cascades.

Rho-GTPase acts as a molecular switch during inflammatory cell migration by cycling between the inactive Rho-GDP and active Rho-GTP forms. It plays a crucial role in actin cytoskeleton dynamics and the precise regulation of leukocyte immune functions. Previous reports have indicated that dysregulation of Rho-GTPase signaling is associated with various inflammatory diseases [29]. Small GTPase can mediate ERK1/2 entry into the nucleus through the MAPK signaling pathway, leading to apoptosis, inflammatory stress responses, and ROS production [30]. ROS act as a double-edged sword, potentially playing a role in both pro-inflammatory and anti-inflammatory processes.

Recent studies have revealed the physiological importance of ROS as crucial signaling molecules for maintaining cellular functions and homeostasis [31]. Infection with *S. agalactiae* in human endothelial cells induces ROS [32], which can persist for a week compared to *Staphylococcus aureus* and *Escherichia coli* [33]. In the present study, we found that genes related to the regulation of metabolic processes of ROS were upregulated as mammary epithelial cells became increasingly infected with the pathogen. This indicates that mammary epithelial cells employ ROS signaling as a defense mechanism during infection. However, ROS production also activates the ERK1/2 pathway, triggering various immune processes to eliminate cells, leading to apoptosis [34,35]. ROS can also induce autophagy, which allows pathogens to survive in mammary epithelial cells [36]. These findings suggest that during *S. agalactiae* infection of mammary epithelial cells, the ERK1/2 pathway induces inflammatory responses through MAPK signaling, accompanied by increased ROS production.

KEGG analysis identified several crucial pathways, including transcription misregulation in cancer, the IL-17 signaling pathway, the P53 signaling pathway, the TGF-beta signaling pathway, the MAPK signaling pathway, pathways in cancer, and apoptosis. Transcription misregulation in cancer pathways was triggered by pathogen invasion and interferes with the regulation of cancer-related transcription factors. Upregulated *RUNX1* inhibits the invasiveness of most breast cancer subtypes, especially in the early stages of tumorigenesis, and prevents the epithelial–mesenchymal transition in breast cancer cells [32]. In the time-series analysis, the expression of *RUNX1* gradually increased with the severity of bacterial infection, probably due to bacterial interference and the disruption of the cellular regulatory system. Recent studies have shown that the IL-17 signaling pathway is also involved in the occurrence of mastitis [37] and is particularly related to the inflammatory response in mammary epithelial cells [38]. The p53 and MAPK signaling pathways were enriched in exosomes from cells infected with bacteria [39]. The p53 signaling pathway is a complex cellular stress response network with various inputs and downstream outputs related to its role as a tumor suppressor pathway [40]. The MAPK signaling pathway is involved in tumor formation, invasion, metastasis, and apoptosis [41], and its activation is also involved in mastitis [42]. The TGF-beta signaling pathway induces apoptosis in mammary epithelial cells [43]. Furthermore, TGF-beta1 can cooperate with the ERK1/2 pathway to promote Gram-positive bacterial adhesion and infection of mammary epithelial cells [44]. Additionally, the RhoA/Rho kinase signaling cascade aids in changes induced by TGF-beta in cytoskeletal organization and cell permeability [45]. *BCL2L11*, which is involved in the apoptotic pathway, participates primarily in the extrinsic apoptotic signaling pathway in the absence of ligands, the positive regulation of cysteine-type endopeptidase activity involved in the apoptotic process, and protein kinase binding between upregulated genes. It is also involved in resistance to EGFR tyrosine kinase inhibitors. 

It is possible that, upon interference by pathogens, host cells initiate immune and inflammatory responses to combat bacterial infection, potentially leading to adverse effects such as progression toward cancer. The enriched signaling pathways in the upregulated Clusters 1 and 3 suggest that cells may gradually transform toward a cancerous state. However, these cells did not evade apoptosis. In Cluster 3, several disease-related pathways were upregulated, including those involved in cancer. Moreover, *BCL2L11* was enriched in this pathway along with other genes.

### 4.2. Alternative Splicing Events and Associated Biological Function Occurring in Mammary Epithelial Cells Following Infection by S. agalactiae ATCC 27956

Alternative splicing is a crucial mechanism of genetic regulation that enhances the diversity and complexity of the transcriptome and proteome of a limited number of genes. Numerous studies have suggested that alternative splicing events lead to changes in protein expression or function during the onset and progression of the disease [46]. In this study, variable splicing showed that DEmRNAs were mainly involved in the regulating of GTPase activity, protein ubiquitination, ATP binding, and RNA binding. Some studies have suggested that immune-related GTPase directly mediates the pathogen membrane by binding and exposing the pathogen to cytoplasmic defenses [47]. Immune-related GTPase can also use ubiquitination to tag intracellular pathogens [48]. In this study, differentially spliced genes were found to be involved in ubiquitin-dependent protein catabolism and protein polyubiquitination. This indicates that infection by *S. agalactiae* of mammary epithelial cells activates the immune defense mechanisms of the host cell, leading to overactivation of the GTPase system, which collaborates with the intracellular ubiquitination system to combat intracellular bacterial damage. However, the influence of intracellular bacterial molecular regulation disrupts normal transcriptional regulation in host cells, particularly the normal alternative splicing process, which could cause dysregulation of GTPase signaling. Differentially spliced genes were enriched in ATP- and RNA-binding functions, indicating that *S. agalactiae* interferes with the splicing of ATP- and RNA-binding proteins, indirectly affecting cellular energy metabolism and transcriptional regulation. 

Among the three groups of DEmRNAs, the upregulated gene *RUNX1* exhibited MXE, A3SS, and SE events in the S_M group (not significant); A3SS and SE events in the H_M group (not significant); and significant A3SS and SE events in the H_S group. KEGG analysis of the H_S group indicated that *RUNX1* is primarily involved in cancer-related pathways, including cancer transcription misregulation pathways and tight junctions. GO analysis revealed the involvement of nucleoplasmic localization, ATP binding, and protein-containing complex processes. This suggests that as the bacterial infection intensifies, the *RUNX1* splicing process in host cells is disrupted, altering its expression pattern and potentially increasing the risk of cell transformation. Studies have shown that *RUNX1* is associated with RNA Pol II-transcribed proteins, lncRNA genes, and RNA Pol I-transcribed ribosomal genes, which are crucial for the growth and maintenance of the mammary epithelial cell phenotype [49]. This further implies that bacterial interference with transcription indirectly induces morphological changes in mammary epithelial cells, contributing to carcinogenesis. Recent studies have indicated that *RUNX1* plays a role in breast cancer cell migration and invasion [50].

### 4.3. Impact of Alternative Splicing Events on Potential Candidate Genes

In this study, through analysis of DElncRNA target gene prediction, alternatively spliced genes, and DEmRNAs, we identified eight common genes, including *RUNX1* and *BCL2L11*. *RUNX1* is a crucial transcription factor that induces the expression of several genes. Among the upregulated genes, *BCL2L11* induces the expression of *RUNX1* [51]. *BCL2L11* plays a dual role in the mechanisms of disease by inhibiting autophagy and initiating apoptosis [52]. Under normal conditions, *BCL2L11* undergoes alternative splicing to produce at least 18 different isoforms [51]. However, in this study, however, *BCL2L11* underwent an MXE event in the H and M groups. MXE results in different exon combinations that may maintain protein folding but alter the specificity and selectivity of protein function [53]. This indicates that under continuous bacterial infection, alternative splicing of *BCL2L11* is affected, altering its splicing form and resulting in changes in *BCL2L11* conformation. This splicing pattern reduces proteomic diversity, leading to protein dysfunction and altered biological functions.

### 4.4. Interaction between Potential Candidate Genes and Differentially Expressed S. agalactiae ATCC 27956 Genes

In the time-series analysis, genes involved in BPs of DNA-binding transcription activator activity were upregulated in Clusters 1 and 6. However, the transcription factors of the host cells were negatively regulated in Clusters 5 and 8. This indicates that some pathogens factors have replaced host transcription factors, thus affecting host transcriptional regulation. It has been hypothesized that during pathogen infection of host cells, the host’s spliceosome is disrupted and consequently downregulated. The host cell nucleus is similarly affected, which interferes with the host’s transcriptional regulation, particularly the activation of nucleic acid transcription factors and the specificity of RNA polymerase.

Interestingly, in the GO enrichment analysis of DEmRNAs in the host cell transcriptome, both the spliceosome and mRNA splicing processes were downregulated, indicating that bacterial interference affects the normal gene splicing process in the mammary epithelial cells. Conversely, some proteins secreted by *S. agalactiae* were also involved. Combined with the differential expression analysis of pathogenic bacteria, multiple genes were upregulated during infection, suggesting an influence on the transcriptional regulation of host cells and their interactions during infection. For example, *nusB*, *rimN*, *yhbY*, *infC*, *prfB*, and *tsf* were involved. Studies have shown that *nusB* is transcribed in contrast to the eukaryotic system and may be a potential antibacterial target [54]. In additionally, biological coupling of transcription and translation control downstream gene expression [55]. The YrdC protein translated from *rimN* is involved in tRNA modification and preferentially binds to RNA [56,57]. *yhbY* is involved in ribosomal assembly and exhibits RNA-binding activity [58]. *infC* guides transcriptional regulation and is upregulated during bacterial infection of the animal liver [59,60]. The RF2 protein encoded by *prfB* is required for the recognition of stop codons during the termination of bacterial translation [61]. This suggests that bacteria proliferate extensively and express proteins capable of invading host cells. *tsf* can contribute to the production of biologically active bacterial keratinases [62], and this site can confer strong antibiotic resistance [63], making it a potential therapeutic target. To adapt to the host system, bacteria employ various strategies, including the production of virulence factors and the formation of biofilms, to escape the host immune system and resist antibiotics [64]. 

Bacteria can manipulate host signaling pathways by regulating host lncRNAs to escape immune clearance. Therefore, bacteria can induce significant alteration in the cell transcriptome and develop various strategies to modify immune signaling for its survival [65]. Currently, lncRNAs have been shown to play crucial roles in the regulation of alternative splicing in response to various stimuli or diseases [66]. Furthermore, increasing evidence indicates that lncRNAs are important in regulatory circuits that control innate and adaptive immune responses to bacterial pathogens [67]. In the analysis of the co-expression network, three bacterial genes (*tsf*, *prfB,* and *infC*) had the most significant effect on the differential targeting of lncRNAs to *RUNX1* and *BCL2L11*. Both *RUNX1* and *BCL2L11* undergo abnormal alternative splicing during infection. It is hypothesized that during infection of bovine mammary epithelial cells by *S. agalactiae*, genes such as *tsf*, *prfB*, and *infC*, which are involved in RNA binding, infiltration of host cells, and disruption of lncRNA targeting of *RUNX1* and *BCL2L11*. This affects the normal alternative splicing process of the host, disrupting the regulation of normal cell proliferation and apoptosis.

## 5. Conclusions

In this study, we analyzed the infection of bovine mammary epithelial cells with *S. agalactiae ATCC 27956* using absolute quantitative interaction transcriptome sequencing. Analysis of the results revealed that when *S. agalactiae* infection triggers both immune and inflammatory responses in mammary epithelial cells, it also induces cell carcinogenesis and apoptosis. Furthermore, to evade cellular immune defenses, *S. agalactiae* interferes with normal alternative splicing processes by generating lncRNAs that disrupt the regulation of apoptosis and disease-related pathways, thereby achieving immune evasion.

## Figures and Tables

**Figure 1 animals-14-02587-f001:**
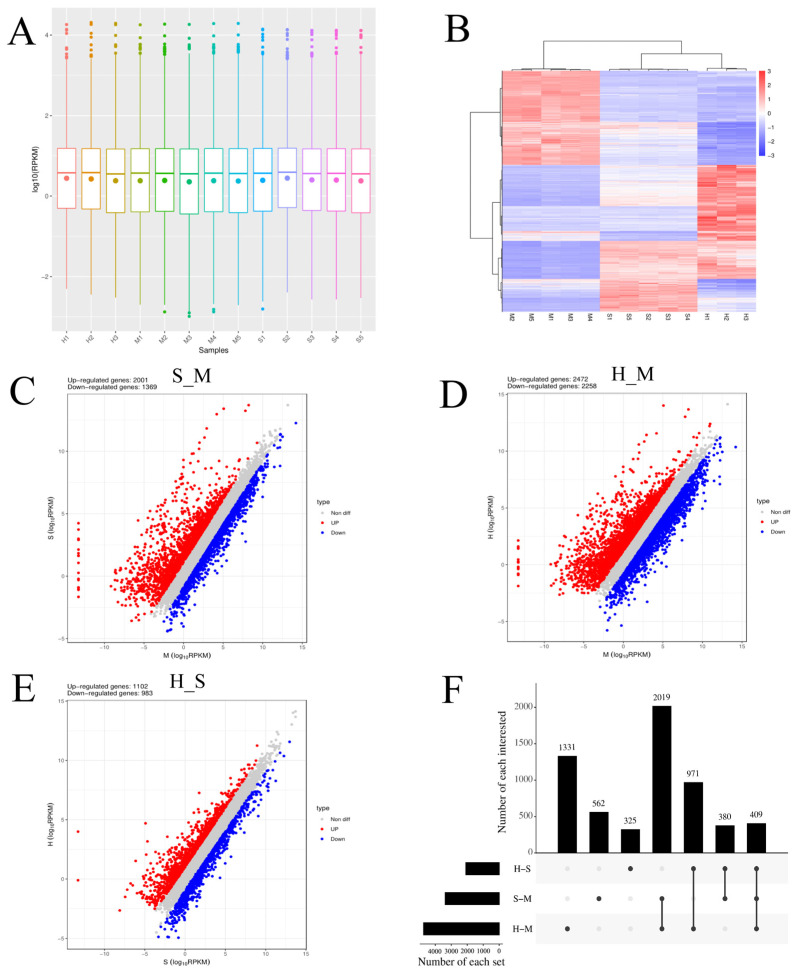
Screening differently expressed mRNAs (DEmRNAs) in *S. agalactiae ATCC 27956* infected mammary epithelial cells among the control (*n* = 5), *S. agalactiae*—S groups (*n* = 5), and *S. agalactiae*—H groups (*n* = 3). (**A**) Gene expression level analysis in S_M, H_M, and H_S. The *X*-axis of the box plot represents the sample name, while the *Y*-axis represents log10 (RPKM). The box plots for each region correspond to five statistical measures (maximum, upper quartile, median, lower quartile, and minimum values, respectively). (**B**) Cluster analysis of DEmRNAs in mammary epithelial cells between the control group (M1, M2, M3, M4, and M5), S groups (S1, S2, S3, S4, and S5), and H-treated groups (H1, H2, and H3). Red indicates highly expressed genes, and blue indicates low expressed genes. Each column represents a sample, and each row represents a gene. On the left is the tree diagram of mRNA clustering. The closer the two mRNA branches are, the closer their expression level is. The upper part is the tree diagram of sample clustering, and the bottom is the name of each sample. The closer the two-sample branches are to each other, the closer the expression pattern of all genes in the two samples is and the trend of the more recent gene expression. (**C**–**E**) Volcano plot of global DEmRNAs in S_M, H_M, and H_S, respectively. Red dots (up) represent significantly upregulated genes (*p*-values < 0.05, log2(fold-change) > 1); blue dots (down) represent significantly downregulated genes (*p*-values < 0.05, log2(fold-change) < −1); gray dots represent insignificantly differential expressed genes. (**F**) Upset map analysis of S_M, H_M, and H_S. The origin and connecting lines of the *X*-axis represent intersections, while the black bars represent the number of differentially expressed genes in each group. The number of differentially expressed genes at the intersection of each group on the *Y*-axis.

**Figure 2 animals-14-02587-f002:**
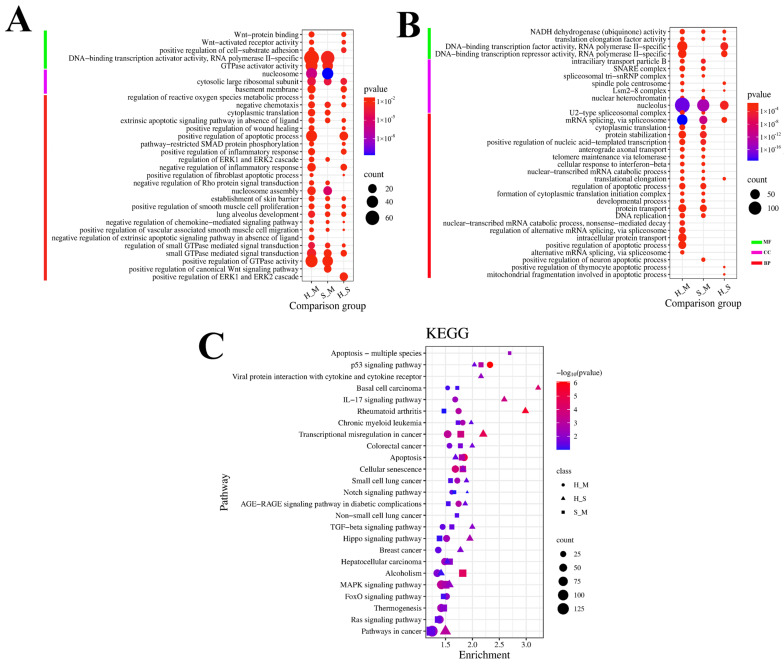
GO and KEGG analysis of DEmRNAs in S_M, H_M, and H_S. (**A**) The *Y*-axis on the left represents GO terms of upregulated genes, including biological process (BP), cellular component (CP), and molecular function (MF). The *X*-axis indicates different comparison groups. The area of a circle represents the DEG number. Low *p*-values are shown in the red circle, and high *p*-values are shown in the blue circle. (**B**) The *Y*-axis on the left represents GO terms of downregulated genes, including biological process (BP), cellular component (CP), and molecular function (MF). The *X*-axis indicates different comparison groups. The area of a circle represents the DEG number. Low *p*-values are shown in the red circle, and high *p*-values are shown in the blue circle. (**C**) The *Y*-axis on the left represents KEGG pathways, and the *X*-axis indicates the gene enrichment of each term. The shapes represent different groups. The area of shapes represents DEmRNA numbers.

**Figure 3 animals-14-02587-f003:**
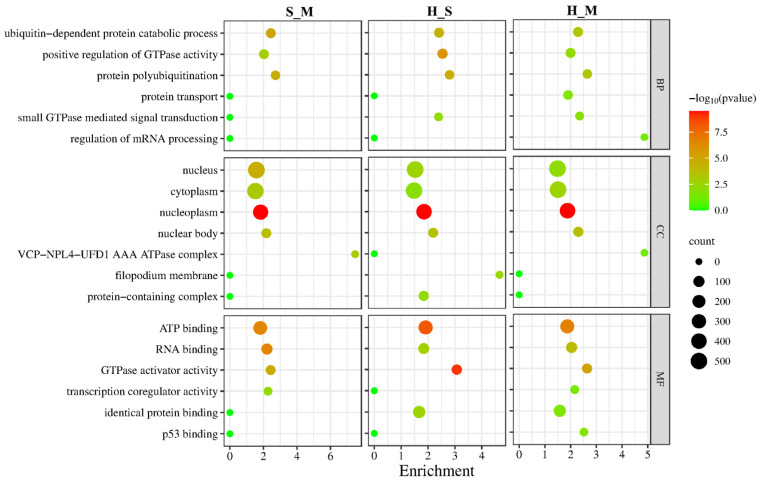
GO enrichment results of differentially spliced genes in S_M, H_M, and H_S. The *Y*-axis on the left represents GO terms, including biological process (BP), cellular component (CP), and molecular function (MF), and the *X*-axis indicates gene enrichment of each term. Low *p*-values are shown in the red circle, and high *p*-values are shown in the green circle. The area of a circle represents the DEmRNA number.

**Figure 4 animals-14-02587-f004:**
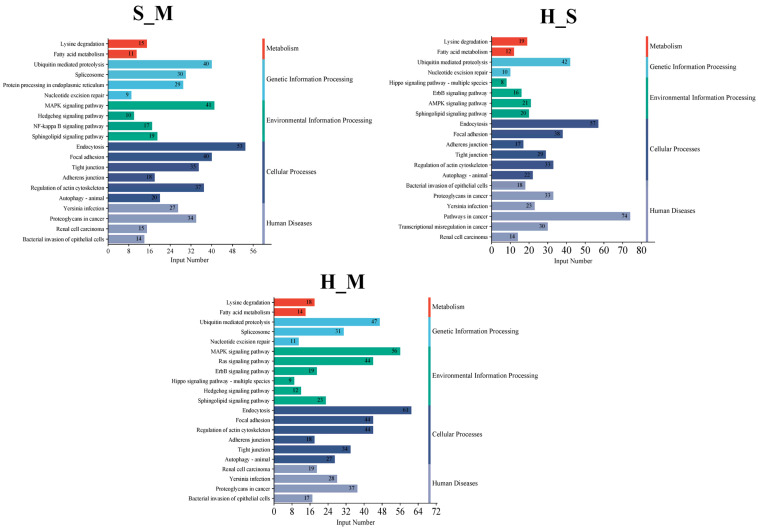
The pathways of spliced genes in S_M, H_M, and H_S. The *Y*-axis on the left represents KEGG pathways, the *Y*-axis on the right represents the major category to which each pathway belongs, and the *X*-axis indicates the DEmRNA numbers of each pathway.

**Figure 5 animals-14-02587-f005:**
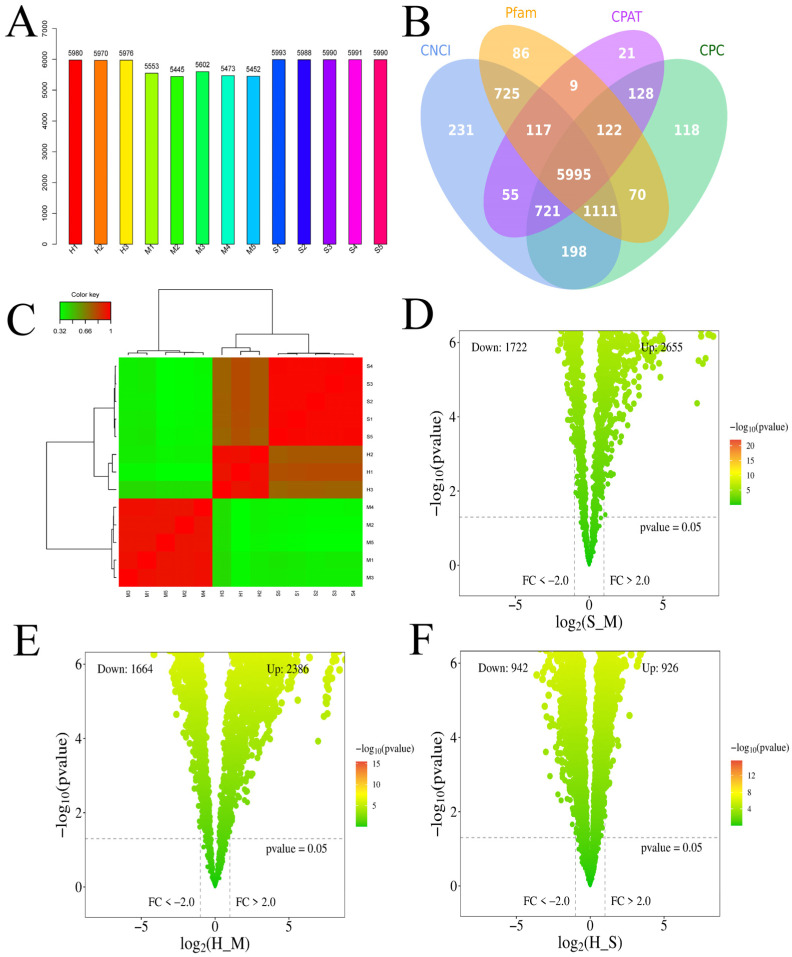
Screening DElncRNAs compared between the M group, S group, and H group. (**A**) Distribution of DElncRNAs in each sample, with the *Y*-axis representing the number of genes and the *X*-axis representing different samples; (**B**) Venn analysis of novel DElncRNAs obtained from four software programs: CNCI, CPC, Pfam, and CPAT. (**C**) Cluster analysis of DElncRNAs in mammary epithelial cells between the control group (M1, M2, M3, M4, and M5), normally treated groups (S1, S2, S3, S4, and S5), and deeply treated groups (H1, H2, and H3). Red indicates highly expressed genes, and green indicates low expressed genes. Each column represents a sample. (**D**–**F**) Volcano plot of global DElncRNAs in S_M, H_M, and H_S, respectively. Gradient red dots represent significantly regulated genes (*p* < 0.05, |log2(fold-change)| > 1); dark green dots represent significantly differential expressed genes.

**Figure 6 animals-14-02587-f006:**
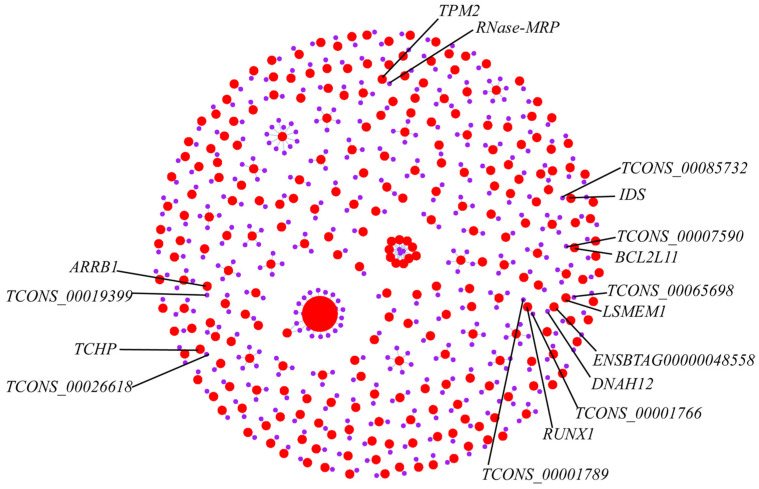
Prediction of target genes for DElncRNA. The red circle represents lncRNA, and the purple circle represents mRNA. The area of the circle represents the degree of connectivity between genes.

**Figure 7 animals-14-02587-f007:**
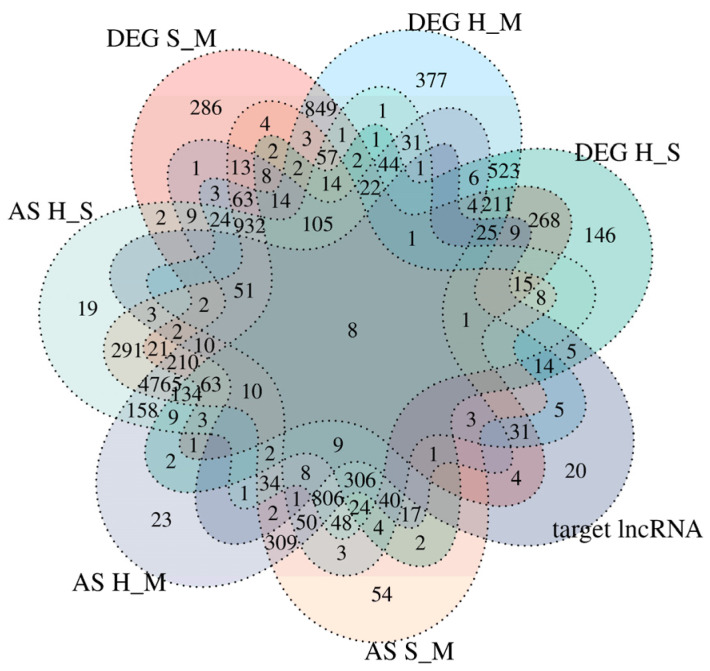
Venn map analysis of DElncRNA target genes, DEmRNA, and AS genes. DEG means differential expression genes; AS means alternative splicing; S_M, H_S, and H_M represent three comparison groups; target lncRNA means three comparative groups targeting mRNA.

**Figure 8 animals-14-02587-f008:**
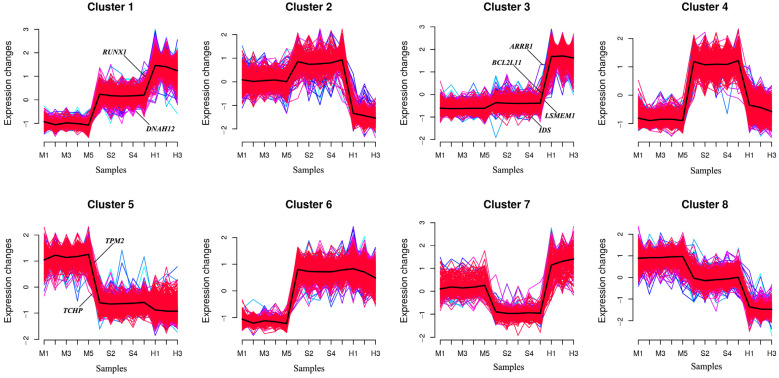
Trend analysis of *S. agalactiae* infection in breast epithelial cells. This series of charts uses Mfuzz to illustrate the dynamic changes in DEmRNAs during pathogen infection. In eight clusters, pink, light blue and blue lines all represent genes with large expression amplitudes. The red area represents genes with similar expression trends. The black line represents the expression trend of the cluster.

**Figure 9 animals-14-02587-f009:**
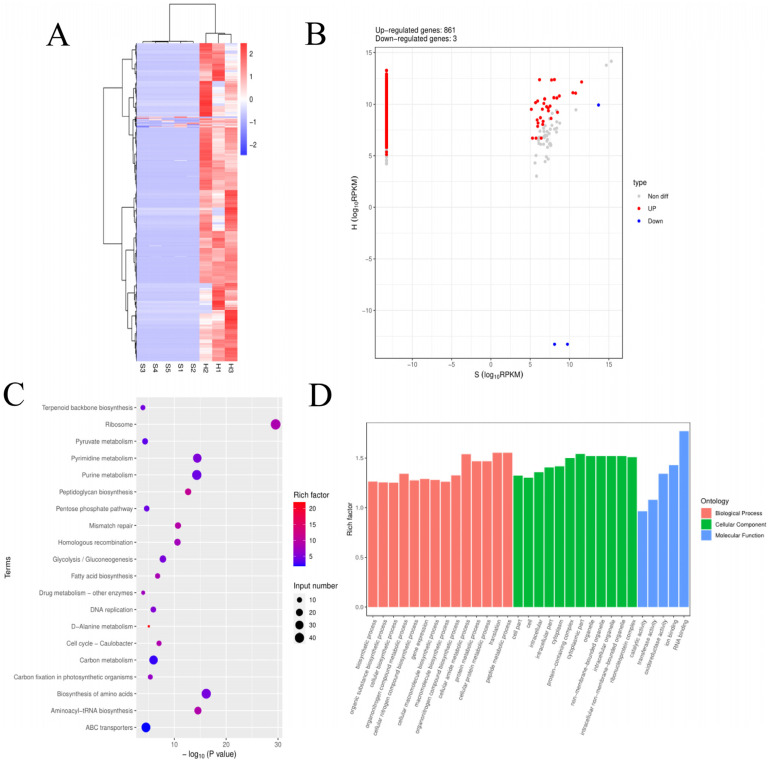
Screening and enrichment analysis of pDEmRNAs of *S. agalactiae ATCC 27956* normally treated groups (*n* = 5) compared with *S. agalactiae* deeply treated groups. (**A**) Cluster analysis of pDEmRNAs in *S. agalactiae* between normally treated groups (S1, S2, S3, S4, and S5) and deeply treated groups (H1, H2, and H3). Red indicates highly expressed genes, and blue indicates low expressed genes. Each column represents a sample, and each row represents a gene. On the left is the tree diagram of mRNA clustering. (**B**) Volcano plot of global pDEmRNAs in *S. agalactiae* between normally treated groups and deeply treated groups. Red dots (up) represent significantly upregulated genes (*p* < 0.05, log2(fold-change) > 1); blue dots (down) represent significantly downregulated genes (*p* < 0.05, log2(fold-change) < −1); gray dots represent insignificantly differential expressed genes. (**C**) KEGG pathway classified annotation of pDEmRNAs in *S. agalactiae*. The pathway is exhibited on the left axis, and the area of the circle represents the number of genes listed on the right axis. (**D**) Annotation of pDEmRNAs using Gene Ontology (GO) in *S. agalactiae*. The rich factor of mRNAs for each GO annotation is exhibited on the left axis.

**Figure 10 animals-14-02587-f010:**
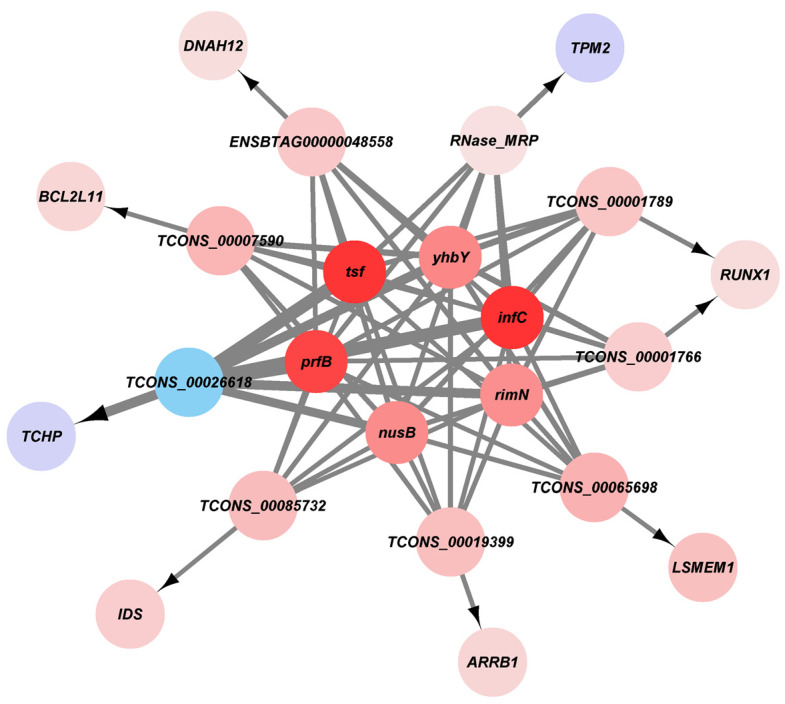
Co-expression network of host cell DElncRNA, DEmRNA, and pathogen pDEmRNA. Red indicates upregulation, blue indicates downregulation, and the color intensity represents strength.

**Figure 11 animals-14-02587-f011:**
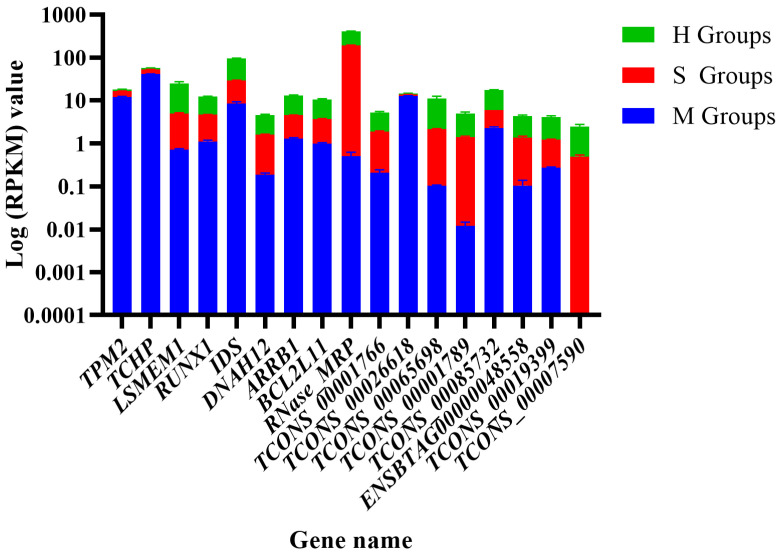
The expression levels of candidate genes.

**Table 1 animals-14-02587-t001:** mRNA sequence quality.

Sample	Group	Total Raw Reads	Total Clean Reads	Total Clean Base (G)	Effective Rate (%)	Reads with UIDs	Dedup Reads
M1	Control (M Group)	81,298,832	70,041,960	10.38	86.15	64,794,252 (92.51%)	61,112,168 (87.25%)
M2		80,917,920	70,388,474	10.46	86.99	65,098,992 (92.49%)	60,322,418 (85.70%)
M3		92,091,998	79,660,982	11.81	86.50	73,798,986 (92.64%)	69,202,752 (86.87%)
M4		82,288,064	70,524,828	10.46	85.70	65,226,928 (92.49%)	61,014,132 (86.51%)
M5		91,369,374	79,413,762	11.81	86.92	73,703,106 (92.81%)	67,255,788 (84.69%)
S1	Treat1 (S Group)	102,946,650	92,613,194	13.63	89.96	86,948,336 (93.88%)	79,863,190 (86.23%)
S2		71,198,348	63,027,196	9.25	88.52	59,120,464 (93.80%)	56,144,004 (89.08%)
S3		86,815,548	76,934,788	11.25	88.62	72,256,950 (93.92%)	68,365,948 (88.86%)
S4		92,741,594	82,904,518	12.12	89.39	77,832,968 (93.88%)	73,052,626 (88.12%)
S5		104,732,530	93,522,804	13.67	89.30	87,853,102 (93.94%)	82,741,910 (88.47%)
H1	Treat2 (H Group)	60,552,640	43,447,378	6.30	72.94	41,488,876 (95.49%)	40,567,390 (93.37%)
H2		72,552,826	51,713,480	7.61	71.28	49,400,638 (95.53%)	46,882,558 (90.66%)
H3		83,335,490	62,442,120	9.20	74.93	59,544,572 (95.36%)	55,991,706 (89.67%)

**Table 2 animals-14-02587-t002:** Type of alternative splicing and statistics of differential alternative splicing events.

EventType.	NumEvents. JC. Only			SigEvents. JC. Only (Up:Down)			NumEvents. JC+ Reads On Target			SigEvents. JC+ Reads on Target (Up:Down)		
	S_M	H_M	H_S	S_M	H_M	H_S	S_M	H_M	H_S	S_M	H_M	H_S
SE	36,032	33,110	35,535	428:497	557:1034	434:818	36,038	33,113	35,536	450:533	593:1091	462:851
MXE	7816	6661	7424	993:1090	1398:1132	1159:794	7816	6661	7424	982:1083	1375:1130	1144:787
A5SS	348	324	309	17:16	19:21	9:12	349	324	309	17:15	21:23	13:13
A3SS	418	405	393	12:12	18:13	12:10	418	405	393	12:13	19:13	12:09
RI	494	455	426	7:21	13:37	8:17	503	457	432	6:18	13:32	7:13

SE, skipped exon; A5SS, alternative 5’splice; A3SS, alternative 3’splice; MXE, mutually exclusive; RI, retained intron.

## Data Availability

The datasets presented in this study can be found in online repositories. The names of the repository/repositories and accession numbers can be found at https://www.ncbi.nlm.nih.gov/sra/?term=PRJNA1151263 (accessed on 23 August 2024).

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
