# Peer review of "UID-Dual Transcriptome Sequencing Analysis of the Molecular Interactions between Streptococcus agalactiae ATCC 27956 and Mammary Epithelial Cells"

_animals, 2024, doi:10.3390/ani14172587_

Round 1

Reviewer 1 Report

Comments and Suggestions for Authors

Dear Editor,

The paper titled " UID- Dual seq Analysis of the Molecular Interactions Between Streptococcus. agalactiae and Mammary Epithelial Cells" explores a compelling and relevant topic, and the authors demonstrate a clear understanding of the research area. The discussion is well-developed and effectively outlines the research scope. However, to reach its full potential, the manuscript would benefit from substantial revisions to enhance clarity and overall quality.

Figure panels 2 and 4 are too small to comprehend and it's not easy to read.

The discussion section offers valuable insights. However, the presentation of technical details could be strengthened by a more logical and coherent structure. I recommend a careful reorganization of this content to improve clarity and flow.

To elevate the manuscript, I recommend the methodology for RNA seq and downstream analysis for splicing and pathway analysis are expanded to provide more details for readers to reproduce and the submission of the raw sequencing data sets to public data repositories. 

 With these revisions, the paper has the potential to make a significant contribution to the field.

Sincerely

Comments on the Quality of English Language

Specific comments: While the ideas presented are promising, the text would be significantly improved with a careful review for grammatical accuracy . Several instances of grammatical errors and redundant phrasing detract from the clarity of the message. As an example, in  Line 152 : instead of PE150 mode the authors spelled it as 'PE150 model' and in  Line 198 : the  word figure 1 has been repeated twice.  Additionally, some sentences are overly complex. Breaking these down into shorter, more concise statements would enhance readability.

Author Response

Dear editors and reviewers,

      Thank you for your letter and for editor and reviewers’ comments concerning our manuscript entitled “UID- Dual seq Analysis of the Molecular Interactions Between Streptococcus. agalactiae and Mammary Epithelial Cells”. Those comments are all valuable and very helpful for revising and improving our manuscripts, as well as the important guiding significance to our research. We have studied comments carefully and have made corrections which we hope meet with approval. Revised portions are marked in red on the manuscript. The main corrections in the manuscript and the response to the reviewer’s comments, please see the attachment.

        We feel great thanks for your professional review work on our manuscript.

Reviewer 2 Report

Comments and Suggestions for Authors

Dear authors

This manuscript is relatively well written with exception in the introduction and discussion sections.

My comments as below and highlighted yellow in the attached PDF file that I had access to.

I hope my comments are taken as a constructive criticism only.

General comments

1. You investigated the inflammatory changes related to a single ATCC strain (ATCC 27956).  Hence, this should be stated in the title, figure and table captions, and the start of the discussion.

2. The use of a single strain does not guarantee representativeness to the species.  This limitation should be recognized in the discussion.  The recognition of the limitation does not decrease the value of the manuscript.  Instead, it make the research group more self-reflective and able to recognize study limitations that is a characteristic of a positive scientific mind.

Specific comments

L 14 Gram - the first letter does not need capitalization

L18 Please delete 'It was found'

L18 'becomes'

L33 Please delete 'However,'

L41 'in dairy cows'

L42/43 'collected into the milk ducts' is appropriate for duct storing species (e.g., goat and human). Dairy cows are mainly alveoli storing species.  Hence, the milk is mainly stored in the alveoli between milking and oxytocin release is essential for 'squeezing' the alveoli and milk let-down.  Please correct this and find appropriate references (e.g., something from Peaker or Knight).

L43 'Typically'

L46 'disease' is actually an incorrect term.  The correct term is a syndrome.

L47/48 Covers very well the environmental mastitis.  In your manuscript, you discuss about contagious mastitis.  Hence, this statement is not relevant at all. Something in lines: ‘Mastitis syndrome is caused by a variety of microorganisms, predominantly bacteria, split into contagious and environmental.  The source of contagious bacteria are infected quarters and cows.  The source of environmental bacteria is the environment.’

L48 ‘symptom’ is again another wrong term.  Symptom is the verbally expressed feeling by the patient.  As we do not speak cow, we can detect signs not symptoms.

L54 SCC not SSC

L56 ‘may’

L58 ‘common contagious pathogens’

L60 ‘may be’

L67 ‘is’

L132 Please be consistent throughout the manuscript in expressing degrees Celsius (with space or no space between the number and the sing for a degree)

L164 ‘F’

L169 Please delete the full stop

L194 space, please

Table 1 – Please make all values with a same number of decimal spaces

L466 ‘may’ – Authors have ignored the fact that the most common form of S. agalactiae mastitis is actually the chronic subclinical form leading to mammary gland fibrosis and loss of productivity.

Comments on the Quality of English Language

See editors comments

Author Response

(The authors gave the same response as above.)

Reviewer 3 Report

Comments and Suggestions for Authors

The manuscript by Gong and co-authors titled “UID- Dual seq Analysis of the Molecular Interactions Between Streptococcus. agalactiae and Mammary Epithelial Cells” is devoted to analyzing the expression of Streptococcus agalactiae in the eukaryotic cells of cow mammary glands. The manuscript is quite interesting; however, it has one significant drawback that prevents its acceptance. According to the Instructions for Authors, the expression analysis data must be publicly available in the GEO or SRA databases, with the accession number mentioned in the text. The exact quote is the next one: "New high throughput sequencing (HTS) datasets (RNA-seq, ChIP-Seq, degradome analysis, …) MUST be deposited either in the GEO database or in the NCBI’s Sequence Read Archive (SRA)." (Link: https://www.mdpi.com/journal/animals/instructions#suppmaterials). I strongly recommend complying with this journal requirement.

Additionally, it would be beneficial to work on the quality of the English language.

Below are some other minor issues I noted:

- Lines 2 and 27: It's better to write the title fully as it is done below: UID-Dual transcriptome sequencing.

- Line 3: Remove the period after the genus name.

- Line 5: At least the city, province, and country must be added.

- Line 32: Duplicate "to produce"

- Line 38: "Streptococcus agalactiae" should be italicized.

- Line 43: Typo "Tyoically"

- Line 54: Should "somatic cell count" be SCC, not SSC?

- Lines 106, 109: What is the correct name of the method, UID-Dual seq or Dual RNA-seq?

- Lines 117, 118, 122: Add the manufacturer of the media.

- Lines 123, 132: Write "2" in CO2 in lowercase.

- Line 128: Remove the extra "group"

- Lines 139-149: The catalog numbers can be removed. The trademark TM should be in uppercase.

- Line 164: Begin the sentence with a capital letter.

- Line 170: Remove the period at the end of the link.

- Line 176: Typo "determine"

- Lines 177-189: These can be combined, as having sections of one and two lines appears strange.

- Line 198 and onwards: Figures and tables should be placed after their first mention in the text, not before.

- Line 302: "Yersinia" should be italicized.

- Lines 311, 318: There are no mentions of these databases in the materials and methods. Are these programs (line 311) or databases (line 318)? Is it correct CNKI or CNCI?

- Line 329: Could you elaborate on what "analysis" means? Where are the labels? What does this figure represent? It's explained very briefly.

- Lines 342, 345: There is nothing about Mfuzz/Mfuzzy in the materials and methods. What is the correct form, Mfuzz or Mfuzzy? What version of Mfuzzy is this?

- Line 386: Full meanings of MF etc. should be given at their first mention, not randomly.

- Figure 10: What do the different sizes of the circles mean?

- Line 646: Add an explanation of what figures A, B, C, etc. represent.

- Line 665: As mentioned above, there should be an accession number in the GEO or SRA database here.

Comments on the Quality of English Language

It would be beneficial to work on the quality of the English language.

Author Response

(The authors gave the same response as above.)

Round 2

Reviewer 2 Report

Comments and Suggestions for Authors

acceptable

Comments on the Quality of English Language

-

Reviewer 3 Report

Comments and Suggestions for Authors

The authors have addressed all the comments.

I have no further complaints about the manuscript.

Comments on the Quality of English Language

The English language is okay.